# Hierarchical Reinforcement Learning by Discovering Intrinsic Options

**Jesse Zhang**[*][†][1], **Haonan Yu**[*][2], **Wei Xu**[2]
[1]University of Southern California, [2]Horizon Robotics

## Abstract

We propose a hierarchical reinforcement learning method, HIDIO, that can learn task-agnostic options in a self-supervised manner while jointly learning to utilize them to solve sparse-reward tasks. Unlike current hierarchical RL approaches that tend to formulate goal-reaching low-level tasks or pre-define ad hoc lower-level policies, HIDIO encourages lower-level option learning that is independent of the task at hand, requiring few assumptions or little knowledge about the task structure. These options are learned through an intrinsic entropy minimization objective conditioned on the option sub-trajectories. The learned options are diverse and task-agnostic. In experiments on sparse-reward robotic manipulation and navigation tasks, HIDIO achieves higher success rates with greater sample efficiency than regular RL baselines and two state-of-the-art hierarchical RL methods. Code available at https://www.github.com/jesbu1/hidio.

## 1 Introduction

Imagine a wheeled robot learning to kick a soccer ball into a goal with sparse reward supervision. In order to succeed, it must discover how to first navigate in its environment, then touch the ball, and finally kick it into the goal, only receiving a positive reward at the end for completing the task. This is a naturally difficult problem for traditional reinforcement learning (RL) to solve, unless the task has been manually decomposed into temporally extended stages where each stage constitutes a much easier subtask. In this paper we ask, how do we learn to decompose the task automatically and utilize the decomposition to solve sparse reward problems?

Deep RL has made great strides solving a variety of tasks recently, with hierarchical RL (hRL) demonstrating promise in solving such sparse reward tasks (Sharma et al., 2019b; Le et al., 2018; Merel et al., 2019; Ranchod et al., 2015). In hRL, the task is decomposed into a hierarchy of subtasks, where policies at the top of the hierarchy call upon policies below to perform actions to solve their respective subtasks. This abstracts away actions for the policies at the top levels of the hierarchy. hRL makes exploration easier by potentially reducing the number of steps the agent needs to take to explore its state space. Moreover, at higher levels of the hierarchy, temporal abstraction results in more aggressive, multi-step value bootstrapping when temporal-difference (TD) learning is employed. These benefits are critical in sparse reward tasks as they allow an agent to more easily discover reward signals and assign credit.

Many existing hRL methods make assumptions about the task structure (*e.g.*, fetching an object involves three stages: moving towards the object, picking it up, and combing back), and/or the skills needed to solve the task (*e.g.*, pre-programmed motor skills) (Florensa et al., 2016; Riedmiller et al., 2018; Lee et al., 2019; Hausman et al., 2018; Lee et al., 2020; Sohn et al., 2018; Ghavamzadeh & Mahadevan, 2003; Nachum et al., 2018). Thus these methods may require manually designing the correct task decomposition, explicitly formulating the option space, or programming pre-defined options for higher level policies to compose. Instead, we seek to formulate a general method that can learn these abstractions from scratch, for any task, with little manual design in the task domain.

The main contribution of this paper is HIDIO (**HI**erarchical RL by **D**iscovering **I**ntrinsic **O**ptions), a hierarchical method that discovers task-agnostic intrinsic options in a self-supervised manner while

---

[*]Denotes equal contribution. Email to jessez@usc.edu, {haonan.yu,wei.xu}@horizon.ai
[†]Work done as an intern at Horizon Robotics.

learning to schedule them to accomplish environment tasks. The latent option representation is uncovered as the option-conditioned policy is trained, both according to the same self-supervised worker objective. The scheduling of options is simultaneously learned by maximizing environment reward collected by the option-conditioned policy. HIDIO can be easily applied to new sparse-reward tasks by simply re-discovering options. We propose and empirically evaluate various instantiations of the option discovery process, comparing the resulting options with respect to their final task performance. We demonstrate that HIDIO is able to efficiently learn and discover diverse options to be utilized for higher task reward with superior sample efficiency compared to other hierarchical methods.

## 2 PRELIMINARIES

We consider the reinforcement learning (RL) problem in a Markov Decision Process (MDP). Let $\mathbf{s} \in \mathbb{R}^S$ be the agent state. We use the terms "state" and "observation" interchangeably to denote the environment input to the agent. A state can be fully or partially observed. Without loss of generality, we assume a continuous action space $\mathbf{a} \in \mathbb{R}^A$ for the agent. Let $\pi_\theta(\mathbf{a}|\mathbf{s})$ be the policy distribution with learnable parameters $\theta$, and $\mathcal{P}(\mathbf{s}_{t+1}|\mathbf{s}_t, \mathbf{a}_t)$ the transition probability that measures how likely the environment transitions to $\mathbf{s}_{t+1}$ given that the agent samples an action by $\mathbf{a}_t \sim \pi_\theta(\cdot|\mathbf{s}_t)$. After the transition to $\mathbf{s}_{t+1}$, the agent receives a deterministic scalar reward $r(\mathbf{s}_t, \mathbf{a}_t, \mathbf{s}_{t+1})$.

The objective of RL is to maximize the sum of discounted rewards with respect to $\theta$:

$$\mathbb{E}_{\pi_\theta, \mathcal{P}} \left[ \sum_{t=0}^{\infty} \gamma^t r(\mathbf{s}_t, \mathbf{a}_t, \mathbf{s}_{t+1}) \right] \tag{1}$$

where $\gamma \in [0, 1]$ is a discount factor. We will omit $\mathcal{P}$ in the expectation for notational simplicity.

In the options framework (Sutton et al., 1999), the agent can switch between different options during an episode, where an option is translated to a sequence of actions by an option-conditioned policy with a termination condition. A set of options defined over an MDP induces a hierarchy that models temporal abstraction. For a typical two-level hierarchy, a higher-level policy produces options, and the policy at the lower level outputs environment actions conditioned on the proposed options. The expectation in Eq. 1 is taken over policies at both levels.

## 3 HIERARCHICAL RL BY DISCOVERING INTRINSIC OPTIONS

We now introduce our hierarchical method for solving sparse reward tasks. We assume little prior knowledge about the task structure, except that it can be learned through a hierarchy of two levels. The higher-level policy (the *scheduler* $\pi_\theta$), is trained to maximize environment reward, while the lower-level policy (the *worker* $\pi_\phi$) is trained in a self-supervised manner to efficiently discover options that are utilized by $\pi_\theta$ to accomplish tasks. Importantly, by self-supervision the worker gets access to dense intrinsic rewards regardless of the sparsity of the extrinsic rewards.

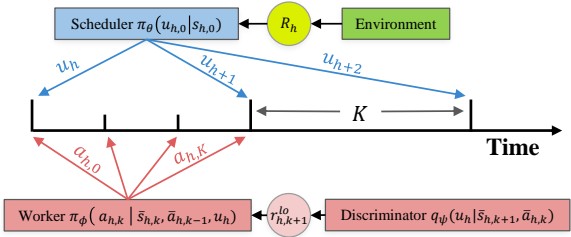

Figure 1: The overall framework of HIDIO. The scheduler $\pi_\theta$ samples an option $\mathbf{u}_h$ every $K$ (3 in this case) time steps, which is used to guide the worker $\pi_\phi$ to directly interact in the environment conditioned on $\mathbf{u}_h$ and the current sub-trajectory $\bar{\mathbf{s}}_{h,k}, \bar{\mathbf{a}}_{h,k-1}$. The scheduler receives accumulated environment rewards $R_h$, while the worker receives intrinsic rewards $r^{\text{lo}}_{h,k+1}$. Refer to Eq. 2 for sampling and Eqs. 3 and 5 for training.

Without loss of generality, we assume that each episode has a length of $T$ and the scheduler outputs an option every $K$ steps. The scheduled option $\mathbf{u} \in [-1, 1]^D$ (where $D$ is a pre-defined dimensionality), is a latent representation that will be learned from scratch given the environment task. Modulated by $\mathbf{u}$, the worker executes $K$ steps before the scheduler outputs the next option. Let the time horizon of the scheduler be $H = \lceil \frac{T}{K} \rceil$. Formally, we define

$$
\begin{array}{llll}
\text{Scheduler policy:} & \mathbf{u}_h \sim \pi_\theta(\cdot|\mathbf{s}_{h,0}), & 0 \leq h < H \\
\text{Worker policy:} & \mathbf{a}_{h,k} \sim \pi_\phi(\cdot|\mathbf{s}_{h,k}, \mathbf{u}_h), & 0 \leq k < K & (2) \\
\text{Environment dynamics:} & \mathbf{s}_{h,k+1} \sim \mathcal{P}(\cdot|\mathbf{s}_{h,k}, \mathbf{a}_{h,k}), & 0 \leq h < H, 0 \leq k < K
\end{array}
$$

where we denote $\mathbf{s}_{h,k}$ and $\mathbf{a}_{h,k}$ as the $k$-th state and action respectively, within the $h$-th option window of length $K$. Note that given this sampling process, we have $\mathbf{s}_{h,K} \equiv \mathbf{s}_{h+1,0}$, namely, the last state of the current option $\mathbf{u}_h$ is the initial state of the next option $\mathbf{u}_{h+1}$. The overall framework of our method is illustrated in Figure 1.

## 3.1 LEARNING THE SCHEDULER

Every time the scheduler issues an option $\mathbf{u}_h$, it receives an reward $R_h$ computed by accumulating environment rewards over the next $K$ steps. Its objective is:

$$
\max_\theta \ \mathbb{E}_{\pi_\theta} \left[ \sum_{h=0}^{H-1} \beta^h R_h \right], \text{where } \beta = \gamma^K \text{ and } R_h = \mathbb{E}_{\pi_\phi} \left[ \sum_{k=0}^{K-1} \gamma^k r(\mathbf{s}_{h,k}, \mathbf{a}_{h,k}, \mathbf{s}_{h,k+1}) \right] \quad (3)
$$

This scheduler objective itself is not a new concept, as similar ones have been adopted by other hRL methods (Vezhnevets et al., 2017; Nachum et al., 2018; Riedmiller et al., 2018). One significant difference between our option with that of prior work is that our option $\mathbf{u}$ is simply a latent variable; there is no explicit constraint on what semantics $\mathbf{u}$ could represent. In contrast, existing methods usually require their options to reside in a subspace of the state space, to be grounded to the environment, or to have known structures, so that the scheduler can compute rewards and termination conditions for the worker. Note that our latent options can be easily re-trained given a new task.

## 3.2 LEARNING THE WORKER

The main focus of this paper is to investigate how to effectively learn the worker policy in a self-supervised manner. Our motivation is that it might be unnecessary to make an option dictate the worker to reach some "$\epsilon$-space" of goals (Vezhnevets et al., 2017; Nachum et al., 2018). As long as the option can be translated to a short sequence of primitive actions, it does not need to be grounded with concrete meanings such as goal reaching. Below we will treat the option as a latent variable that modulates the worker, and propose to learn its latent representation in a hierarchical setting from the environment task.

### 3.2.1 WORKER OBJECTIVE

We first define a new meta MDP on top of the original task MDP so that for any $h$, $k$, and $t$:

1) $\bar{\mathbf{s}}_{h,k} := (\mathbf{s}_{h,0}, \ldots, \mathbf{s}_{h,k})$,
2) $\bar{\mathbf{a}}_{h,k} := (\mathbf{a}_{h,0}, \ldots, \mathbf{a}_{h,k})$,
3) $r(\bar{\mathbf{s}}_{h,k}, \bar{\mathbf{a}}_{h,k}, \bar{\mathbf{s}}_{h,k+1}) := r(\mathbf{s}_{h,k}, \mathbf{a}_{h,k}, \mathbf{s}_{h,k+1})$, and
4) $\mathcal{P}(\bar{\mathbf{s}}_{h,k+1}|\bar{\mathbf{s}}_{h,k}, \bar{\mathbf{a}}_{h,k}) := \mathcal{P}(\mathbf{s}_{h,k+1}|\mathbf{s}_{h,k}, \mathbf{a}_{h,k})$.

This new MDP equips the worker with historical state and action information since the time $(h, 0)$ when an option $h$ was scheduled. Specifically, each state $\bar{\mathbf{s}}_{h,k}$ or action $\bar{\mathbf{a}}_{h,k}$ encodes the history from the beginning $(h, 0)$ up to $(h, k)$ within the option. In the following, we will call pairs $\{\bar{\mathbf{a}}_{h,k}, \bar{\mathbf{s}}_{h,k+1}\}$ option *sub-trajectories*. The worker policy now takes option sub-trajectories as inputs: $\mathbf{a}_{h,k} \sim \pi_\phi(\cdot|\bar{\mathbf{s}}_{h,k}, \bar{\mathbf{a}}_{h,k-1}, \mathbf{u}_h), 0 \leq k < K$, whereas the scheduler policy still operates in the original MDP.

Denote $\sum_{h,k} \equiv \sum_{h=0}^{H-1} \sum_{k=0}^{K-1}$ for simplicity. The worker objective, defined on this new MDP, is to minimize the entropy of the option $\mathbf{u}_h$ conditioned on the option sub-trajectory $\{\bar{\mathbf{a}}_{h,k}, \bar{\mathbf{s}}_{h,k+1}\}$:

$$
\max_\phi \ \mathbb{E}_{\pi_\theta, \pi_\phi} \sum_{h,k} \underbrace{\log p(\mathbf{u}_h|\bar{\mathbf{a}}_{h,k}, \bar{\mathbf{s}}_{h,k+1})}_{\text{negative conditional option entropy}} \underbrace{-\beta \log \pi_\phi(\mathbf{a}_{h,k}|\bar{\mathbf{s}}_{h,k}, \bar{\mathbf{a}}_{h,k-1}, \mathbf{u}_h)}_{\text{worker policy entropy}} \quad (4)
$$

where the expectation is over the current $\pi_\theta$ and $\pi_\phi$ but the maximization is only with respect to $\phi$. Intuitively, the first term suggests that the worker is optimized to confidently identify an option given

a sub-trajectory. However, it alone will not guarantee the diversity of options because potentially even very similar sub-trajectories can be classified into different options if the classification model has a high capacity, in which case we say that the resulting sub-trajectory space has a very high "resolution". As a result, the conditional entropy alone might not be able to generate useful options to be exploited by the scheduler for task solving, because the coverage of the sub-trajectory space is poor. To combat this degenerate solution, we add a second term which maximizes the entropy of the worker policy. Intuitively, while the worker generates identifiable sub-trajectories corresponding to a given option, it should act as randomly as possible to separate sub-trajectories of different options, lowering the "resolution" of the sub-trajectory space to encourage its coverage.

Because directly estimating the posterior $p(\mathbf{u}_h|\bar{\mathbf{a}}_{h,k}, \bar{\mathbf{s}}_{h,k+1})$ is intractable, we approximate it with a parameterized posterior $\log q_\psi(\mathbf{u}_h|\bar{\mathbf{a}}_{h,k}, \bar{\mathbf{s}}_{h,k+1})$ to obtain a lower bound (Barber & Agakov, 2003), where $q_\psi$ is a *discriminator* to be learned. Then we can maximize this lower bound instead:

$$\max_{\phi,\psi} \mathbb{E}_{\pi_\theta, \pi_\phi} \sum_{h,k} \log q_\psi(\mathbf{u}_h|\bar{\mathbf{a}}_{h,k}, \bar{\mathbf{s}}_{h,k+1}) - \beta \log \pi_\phi(\mathbf{a}_{h,k}|\bar{\mathbf{s}}_{h,k}, \bar{\mathbf{a}}_{h,k-1}, \mathbf{u}_h). \tag{5}$$

The discriminator $q_\psi$ is trained by maximizing likelihoods of options given sampled sub-trajectories. The worker $\pi_\phi$ is trained via max-entropy RL (Soft Actor-Critic (SAC) (Haarnoja et al., 2018)) with the intrinsic reward $r_{h,k+1}^{lo} := \log q_\psi(\cdot) - \beta \log \pi_\phi(\cdot)$. $\beta$ is fixed to 0.01 in our experiments.

Note that there are at least four differences between Eq. 5 and the common option discovery objective in either VIC (Gregor et al., 2016) or DIAYN (Eysenbach et al., 2019):

1. Both VIC and DIAYN assume that a sampled option will last through an entire episode, and the option is always sampled at the beginning of an episode. Thus their option trajectories "radiate" from the initial state set. In contrast, our worker policy learns options that initialize every $K$ steps within an episode, and they can have more diverse semantics depending on the various states $s_{h,0}$ visited by the agent. This is especially helpful for some tasks where new options need to be discovered after the agent reaches unseen areas in later stages of training.
2. Actions taken by the worker policy under the current option will have consequences on the next option. This is because the final state $s_{h,K}$ of the current option is defined to be the initial state $s_{h+1,0}$ of the next option. So in general, the worker policy is trained not only to discover diverse options across the current $K$ steps, but also to make the discovery easier in the future steps. In other words, the worker policy needs to solve the credit assignment problem *across* options, under the expectation of the scheduler policy.
3. To enable the worker policy to learn from a discriminator that predicts based on option sub-trajectories $\{\bar{\mathbf{a}}_{h,k}, \bar{\mathbf{s}}_{h,k+1}\}$ instead of solely on individual states $\mathbf{s}_{h,k}$, we have constructed a new meta MDP where each state $\bar{\mathbf{s}}_{h,k}$ encodes history from the beginning $(h,0)$ up to $(h,k)$ within an option $h$. This new meta MDP is critical, because otherwise one simply cannot learn a worker policy from a reward function that is defined by multiple time steps (sub-trajectories) since the learning problem is no longer Markovian.
4. Lastly, thanks to the new MDP, we are able to explore various possible instantiations of the discriminator (see Section 3.3). As observed in the experiments, individual states are actually not the optimal features for identifying options.

These differences constitute the major novelty of our worker objective.

### 3.2.2 SHORTSIGHTED WORKER

It's challenging for the worker to accurately predict values over a long horizon, since its rewards are densely computed by a complex nonlinear function $q_\psi$. Also each option only lasts at most $K$ steps. Thus we set the discount $\eta$ for the worker in two shortsighted ways:

1. `Hard`: setting $\eta = 0$ every $K$-th step and $\eta = 1$ otherwise. Basically this truncates the temporal correlation (gradients) between adjacent options. Its benefit might be faster and easier value learning because the value is bootstrapped over at most $K$ steps ($K \ll T$).
2. `Soft`: $\eta = 1 - \frac{1}{K}$, which considers rewards of roughly $K$ steps ahead. The worker policy still needs to take into account the identification of future option sub-trajectories, but their importance quickly decays.

We will evaluate both versions and compare their performance in Section 4.1.

## 3.3 INSTANTIATING THE DISCRIMINATOR

We explore various ways of instantiating the discriminator $q_\psi$ in order to compute useful intrinsic rewards for the worker. Previous work has utilized individual states (Eysenbach et al., 2019; Jabri et al., 2019) or full observation trajectories (Warde-Farley et al., 2019; Sharma et al., 2019a; Achiam et al., 2018) for option discrimination. Thanks to the newly defined meta MDP, our discriminator is able to take option sub-trajectories instead of current individual states for prediction. In this paper, we investigate six sub-trajectory feature extractors $f_\psi$:

| Feature extractor | Name | Formulation | Explanation |
|---|---|---|---|
| | `State` | $\text{MLP}(\mathbf{s}_{h,k+1})$ | Next state alone |
| | `Action` | $\text{MLP}([\mathbf{s}_{h,0}, \mathbf{a}_{h,k}])$ | Action in context |
| $f_\psi(\bar{\mathbf{a}}_{h,k}, \bar{\mathbf{s}}_{h,k+1}) =$ | `StateDiff` | $\text{MLP}(\mathbf{s}_{h,k+1} - \mathbf{s}_{h,k})$ | Difference between state pairs |
| | `StateAction` | $\text{MLP}([\mathbf{a}_{h,k}, \mathbf{s}_{h,k+1}])$ | Action and next state |
| | `StateConcat` | $\text{MLP}([\bar{\mathbf{s}}_{h,k+1}])$ | Concatenation of states |
| | `ActionConcat` | $\text{MLP}([\mathbf{s}_{h,0}, \bar{\mathbf{a}}_{h,k}])$ | Concatenation of actions |

where the operator $[\cdot]$ denotes concatenation and MLP denotes a multilayer perceptron[1]. Our `State` feature extractor is most similar to DIAYN (Eysenbach et al., 2019), and `StateConcat` is similar to (Warde-Farley et al., 2019; Sharma et al., 2019a; Achiam et al., 2018). However we note that unlike these works, the distribution of our option sub-trajectories is also determined by the scheduler in the context of hRL. The other four feature extractors have not been evaluated before. With the extracted feature, the log-probability of predicting an option is simply computed as the negative squared L2 norm: $\log q_\psi(\mathbf{u}_h | \bar{\mathbf{a}}_{h,k}, \bar{\mathbf{s}}_{h,k+1}) = -\|f_\psi(\bar{\mathbf{a}}_{h,k}, \bar{\mathbf{s}}_{h,k+1}) - \mathbf{u}_h\|_2^2$, by which we implicitly assume the discriminator's output distribution to be a $\mathcal{N}(\mathbf{0}, \mathbf{I_D})$ multivariate Gaussian.

## 3.4 OFF-POLICY TRAINING

The scheduler and worker objectives (Eq. 3 and Eq. 5) are trained jointly. In principle, on-policy training such as A2C (Clemente et al., 2017) is needed due to the interplay between the scheduler and worker. However, to reuse training data and improve sample efficiency, we employ off-policy training (SAC (Haarnoja et al., 2018)) for both objectives with some modifications.

**Modified worker objective** In practice, the expectation over the scheduler $\pi_\theta$ in Eq. 5 is replaced with the expectation over its historical versions. Specifically, we sample options $\mathbf{u}_h$ from a replay buffer, together with sub-trajectories $\{\bar{\mathbf{a}}_{h,k}, \bar{\mathbf{s}}_{h,k+1}\}$. This type of data distribution modification is conventional in off-policy training (Lillicrap et al., 2016).

**Intrinsic reward relabeling** We always recompute the rewards in Eq. 5 using the up-to-date discriminator for every update of $\phi$, which can be trivially done without any additional interaction with the environment.

**Importance correction** The data in the replay buffer was generated by historical worker policies. Thus a sampled option sub-trajectory will be outdated under the same option, causing confusion to the scheduler policy. To resolve this issue, when minimizing the temporal-difference (TD) error between the values of $\mathbf{s}_{h,0}$ and $\mathbf{s}_{h+1,0}$ for the scheduler, an importance ratio can be multiplied: $\prod_{k=0}^{K-1} \frac{\pi_\phi(\mathbf{a}_{h,k} | \bar{\mathbf{s}}_{h,k}, \bar{\mathbf{a}}_{h,k-1}, \mathbf{u}_h)}{\pi_\phi^{old}(\mathbf{a}_{h,k} | \bar{\mathbf{s}}_{h,k}, \bar{\mathbf{a}}_{h,k-1}, \mathbf{u}_h)}$. A similar correction can also be applied to the discriminator loss. However, in practice we find that this ratio has a very high variance and hinders the training. Like the similar observations made in Nachum et al. (2018); Fedus et al. (2020), even without importance correction our method is able to perform well empirically[2].

---

[1]In this paper we focus on non-image observations that can be processed with MLPs, although our method doesn't have any assumption about the observation space.

[2]One possible reason is that the deep RL process is "highly non-stationary anyway, due to changing policies, state distributions and bootstrap targets" (Schaul et al., 2016).

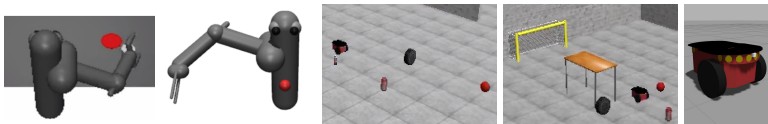

Figure 2: The four tasks we evaluate on. From left to right: 7-DOF PUSHER, 7-DOF REACHER, GOALTASK, and KICKBALL. The first two tasks simulate a one-armed PR2 robot environment while the last two are in the SOCIALROBOT environment. The final picture shows a closeup of the PIONEER2DX robot used in SOCIALROBOT.

## 4 EXPERIMENTS

**Environments** We evaluate success rate and sample efficiency across two environment suites, as shown in Figure 2. Important details are presented here with more information in appendix Section B. The first suite consists of two 7-DOF reaching and pushing environments evaluated in Chua et al. (2018). They both emulate a one-armed PR2 robot. The tasks have sparse rewards: the agent gets a reward of 0 at every timestep where the goal is not achieved, and 1 upon achieved. There is also a small $L_2$ action penalty applied. In 7-DOF REACHER, the goal is achieved when the gripper reaches a 3D goal position. In 7-DOF PUSHER, the goal is to push an object to a 3D goal position. Episodes have a fixed length of 100; a success of an episode is defined to be if the goal is achieved at the *final* step of the episode.

We also propose another suite of environments called SOCIALROBOT [3]. We construct two sparse reward robotic navigation and manipulation tasks, GOALTASK and KICKBALL. In GOALTASK, the agent gets a reward of 1 when it successfully navigates to a goal, -1 if the goal becomes too far, -0.5 every time it is too close to a distractor object, and 0 otherwise. In KICKBALL, the agent receives a reward of 1 for successfully pushing a ball into the goal, 0 otherwise, and has the same distractor object penalty. At the beginning of each episode, both the agent and the ball are spawned randomly. Both environments contain a small $L_2$ action penalty, and terminate an episode upon a success.

**Comparison methods** One baseline algorithm for comparison is standard SAC (Haarnoja et al., 2018), the building block of our hierarchical method. To verify if our worker policy can just be replaced with a naïve action repetition strategy, we compare with SAC+ActRepeat with an action repetition for the same length $K$ as our option interval. We also compare against HIRO (Nachum et al., 2018), a data efficient hierarchical method with importance-based option relabeling, and HiPPO (Li et al., 2020) which trains the lower level and higher level policies together with one unified PPO-based objective. Both are state-of-the-art hierarchical methods proposed to solve sparse reward tasks. Similar to our work, HiPPO makes no assumptions about options, however it utilizes a discrete option space and its options are trained with environment reward.

We implement HIDIO based on an RL framework called ALF [4]. A comprehensive hyperparameter search is performed for every method, with a far greater search space over HiPPO and HIRO than our method HIDIO to ensure maximum fairness in comparison; details are presented in Appendix D.

**Evaluation** For every evaluation point during training, we evaluate the agent with current deterministic policies (by taking $\arg\max$ of action distributions) for a fixed number of episodes and compute the mean success rate. We plot the mean evaluation curve over 3 randomly seeded runs with standard deviations shown as the shaded area around the curve.

### 4.1 WORKER DESIGN CHOICES

We ask and answer questions about the design choices in HIDIO specific to the worker policy $\pi_\phi$.

1. *What sub-trajectory feature results in good option discovery?* We evaluate all six features proposed in Section 3.3 in all four environments. These features are selected to evaluate how different types of subtrajectory information affect option discovery and final performance. They encompass varying types of both local and global subtrajectory information. We plot comparisons of

---

[3] https://github.com/HorizonRobotics/SocialRobot
[4] https://github.com/HorizonRobotics/alf

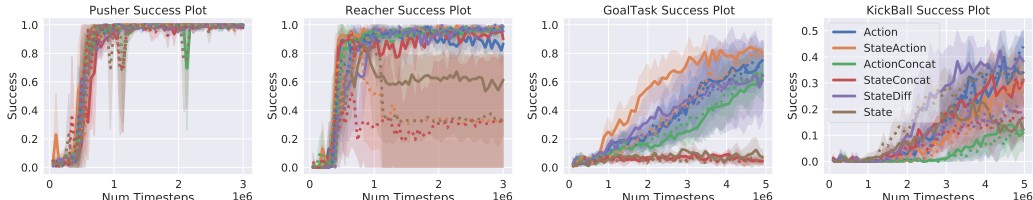

Figure 3: Comparison of all discriminator features against each other across the four environments. Solid lines indicate hard short-sighted workers (`Hard`), dotted lines indicated soft short-sighted workers (`Soft`).

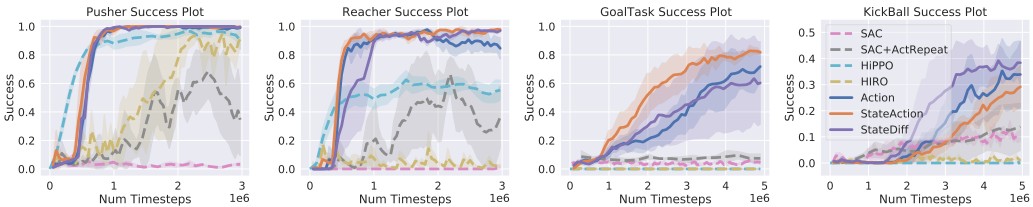

Figure 4: Comparisons of the mean success rates of three features of HIDIO (`Action`, `StateAction`, `StateDiff`; solid lines) against other methods (dashed lines).

sample efficiency and final performance in Figure 3 across all environments (solid lines), finding that `Action`, `StateAction`, and `StateDiff` are generally among the top performers. `StateAction` includes the current action and next state, encouraging $\pi_\phi$ to differentiate its options with different actions even at similar states. Similarly, `Action` includes the option initial state and current action, encouraging option diversity by differentiating between actions conditioned on initial states. Meanwhile `StateDiff` simply encodes the difference between the next and current state, encouraging $\pi_\phi$ to produce options with different state changes at each step.

2. *How do soft shortsighted workers (`Soft`) compare against hard shortsighted workers (`Hard`)?* In Figure 3, we plot all features with `Soft` in dotted lines. We can see that in general there is not much difference in performance between `Hard` and `Soft` except some extra instability of `Soft` in REACHER regarding the `StateConcat` and `State` features. One reason of this similar general performance could be that since our options are very short-term in `Hard`, the scheduler policy has the opportunity of switching to a good option before the current one leads to bad consequences. In a few cases, `Hard` seems better learned, perhaps due to an easier value bootstrapping for the worker.

## 4.2 COMPARISON RESULTS

We compare our three best sub-trajectory features of `Hard`, in Section 4.1, against the SAC baselines and hierarchical RL methods across all four environments in Figure 4. Generally we see that HIDIO (solid lines) achieves greater final performance with superior sample efficiency than the compared methods. Both SAC and SAC+ActRepeat perform poorly across all environments, and all baseline methods perform significantly worse than HIDIO on REACHER, GOALTASK, and KICKBALL.

In PUSHER, HiPPO displays competitive performance, rapidly improving from the start. However, all three HIDIO instantiations achieve nearly 100% success rates while HiPPO is unable to do so. Furthermore, HIRO and SAC+ActRepeat take much longer to start performing well, but never achieve similar success rates as HIDIO. HIDIO is able to solve REACHER while HiPPO achieves only about a 60% success rate at best. Meanwhile, HIRO, SAC+ActRepeat, and SAC are unstable or non-competitive. REACHER is a difficult exploration problem as the arm starts far from the goal position, and we see that HIDIO's automatically discovered options ease exploration for the higher level policy to consistently reach the goal. HIDIO performs well on GOALTASK, achieving 60-80% success rates, while the task is too challenging for every other method. In KICKBALL, the most challenging task, HIDIO achieves 30-40% success rates while every other learns poorly again, highlighting the need for the intrinsic option discovery of HIDIO in these environments.

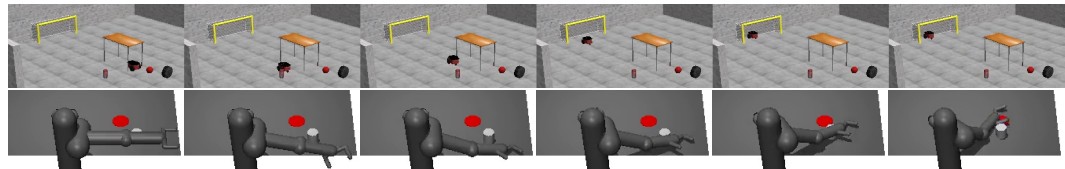

Figure 5: Two example options from the `StateAction` instantiation on KICKBALL (top) and PUSHER (bottom). The top option navigates directly to the goal by bypassing obstructions along the way and the bottom option sweeps the puck towards one direction.

In summary, HIDIO demonstrates greater sample efficiency and final reward gains over all other baseline methods. Regular RL (SAC) fails on all four environments, and while HiPPO is a strong baseline on PUSHER and REACHER, it is still outperformed in both by HIDIO. All other methods fail on GOALTASK and KICKBALL, while HIDIO is able to learn and perform better in both. This demonstrates the importance of the intrinsic, short-term option discovery employed by HIDIO, where the options are diverse enough to be useful for both exploration and task completion.

### 4.3 JOINT $\pi_\phi$ AND $\pi_\theta$ TRAINING

We ask the next question: *is jointly training $\pi_\theta$ and $\pi_\phi$ necessary?* To answer this, we compare HIDIO against a pre-training baseline where we first pre-train $\pi_\phi$, with uniformly sampled options **u** for a portion $\rho$ of total numbers of training time steps, and then fix $\pi_\phi$ while training $\pi_\theta$ for the remaining $(1 - \rho)$ time steps. This is essentially using pre-trained options for downstream higher-level tasks as demonstrated in DIAYN (Eysenbach et al., 2019). We conduct this experiment with the `StateAction` feature on both KICKBALL and PUSHER, with $\rho = \left\{ \frac{1}{16}, \frac{1}{8}, \frac{1}{4} \right\}$. The results are shown in Figure 6. We can see that in

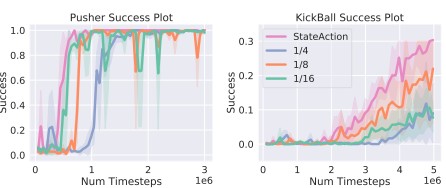

Figure 6: Pretraining baseline comparison at fractions $\left\{ \frac{1}{16}, \frac{1}{8}, \frac{1}{4} \right\}$ of the total number of training time steps.

PUSHER, fewer pre-training time steps are more sample efficient, as the environment is simple and options can be learned from a small amount of samples. The nature of PUSHER also only requires options that can be learned independent of the scheduler policy evolution. Nevertheless, the pre-training baselines seem less stable. In KICKBALL, the optimal pre-training baseline is on $\rho = \frac{1}{8}$ of the total time steps. However without the joint training scheme of HIDIO, the learned options are unable to be used as efficiently for the difficult obstacle avoidance, navigation, and ball manipulation subtasks required for performing well.

### 4.4 OPTION BEHAVIORS

Finally, since options discovered by HIDIO in our sparse reward environments help it achieve superior performance, we ask, *what do useful options look like?* To answer this question, after training, we sample options from the scheduler $\pi_\theta$ to visualize their behaviors in different environments in Figure 5. For each sampled option **u**, we fix it until the end of an episode and use the worker $\pi_\phi$ to output actions given **u**. We can see that the options learned by HIDIO are low-level navigation and manipulation skills useful for the respective environments. We present more visualizations in Figure 9 and more analysis in Section C.2 in the appendix. Furthermore, we present an analysis of task performance for different option lengths in appendix Section C.1 and Figures 7 and 8.

## 5 RELATED WORK

**Hierarchical RL**   Much of the previous work in hRL makes assumptions about the task structure and/or the skills needed to solve the task. While obtaining promising results under specific settings, they may have difficulties with different scenarios. For example, SAC-X (Riedmiller et al., 2018) requires manually designing auxiliary subtasks as skills to solve a given downstream task. SNN4HRL (Florensa et al., 2016) is geared towards tasks with pre-training and downstream components. Lee

et al. (2019; 2020) learns to modulate or compose given primitive skills that are customized for their particular robotics tasks. Ghavamzadeh & Mahadevan (2003) and Sohn et al. (2018) operate under the assumption that tasks can be manually decomposed into subtasks.

The feudal reinforcement learning proposal (Dayan & Hinton, 1993) has inspired another line of works (Vezhnevets et al., 2017; Nachum et al., 2018; Levy et al., 2019; Rafati & Noelle, 2019) which make higher-level manager policies output goals for lower-level worker policies to achieve. Usually the goal space is a subspace of the state space or defined according to the task so that lower-level rewards are easy to compute. This requirement of manually "grounding" goals in the environment poses generalization challenges for tasks that cannot be decomposed into state or goal-reaching.

The MAXQ decomposition (Dietterich, 2000) defines an hRL task decomposition by breaking up the target MDP into a hierarchy of smaller MDPs such that the value function in the target MDP is represented as the sum of the value functions of the smaller ones. This has inspired works that use such decompositions (Mehta et al., 2008; Winder et al., 2020; Li et al., 2017) to learn structured, hierarchical world models or policies to complete target tasks or perform transfer learning. However, building such hierarchies makes these methods limited to MDPs with discrete action spaces.

Our method HIDIO makes few assumptions about the specific task at hand. It follows from the options framework (Sutton et al., 1999), which has recently been applied to continuous domains (Bacon et al., 2017), spawning a diverse set of recent hierarchical options methods (Bagaria & Konidaris, 2020; Klissarov et al., 2017; Riemer et al., 2018; Tiwari & Thomas, 2019; Jain et al., 2018). HIDIO automatically learns intrinsic options that avoids having explicit initiation or termination policies dependent on the task at hand. HiPPO (Li et al., 2020), like HIDIO, also makes no major assumptions about the task, but does not employ self-supervised learning for training the lower-level policy.

**Self-supervised option/skill discovery**    There are also plenty of prior works which attempt to learn skills or options without task reward. DIAYN (Eysenbach et al., 2019) and VIC (Gregor et al., 2016) learn skills by maximizing the mutual information between trajectory states and their corresponding skills. VALOR (Achiam et al., 2018) learns options by maximizing the probability of options given their resulting observation trajectory. DADS (Sharma et al., 2019a) learns skills that are predictable by dynamics models. DISCERN (Warde-Farley et al., 2019) maximizes the mutual information between goal and option termination states to learn a goal-conditioned reward function. Brunskill & Li (2014) learns options in discrete MDPs that are guaranteed to improve a measure of sample complexity. Portable Option Discovery (Topin et al., 2015) discovers options by merging options from source policies to apply to some target domain. Eysenbach et al. (2019); Achiam et al. (2018); Sharma et al. (2019a); Lynch et al. (2020) demonstrate pre-trained options to be useful for hRL. These methods usually pre-train options in an initial stage separate from downstream task learning; few works directly integrate option discovery into a hierarchical setting. For higher dimensional input domains, Lynch et al. (2020) learns options from human-collected robot interaction data for image-based, goal-conditioned tasks, and Chuck et al. (2020) learns a hierarchy of options by discovering objects from environment images and forming options which can manipulate them. HIDIO can also be applied to image-based environments by replacing fully-connected layers with convolutional layers in the early stages of the policy and discriminator networks. However, we leave this to future work to address possible practical challenges arising in this process.

## 6    CONCLUSION

Towards solving difficult sparse reward tasks, we propose a new hierarchical reinforcement learning method, HIDIO, which can learn task-agnostic options in a self-supervised manner and simultaneously learn to utilize them to solve tasks. We evaluate several different instantiations of the discriminator of HIDIO for providing intrinsic rewards for training the lower-level worker policy. We demonstrate the effectiveness of HIDIO compared against other reinforcement learning methods in achieving high rewards with better sample efficiency across a variety of robotic navigation and manipulation tasks.

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

## A  PSEUDO CODE FOR HIDIO

---

**Algorithm 1:** Hierarchical RL with Intrinsic Options Discovery

---

**Input:**

| | | | | | |
|---|---|---|---|---|---|
| $T$ | Episode length | $M$ | Batches per iteration | $\pi_\phi(\mathbf{a}_{h,k}|\mathbf{s}_{h,k},\mathbf{u}_h)$ | Worker |
| $B$ | Batch size | $\alpha$ | Learning rate | $q_\psi(\mathbf{u}_h|\overline{\mathbf{a}}_{h,k},\overline{\mathbf{s}}_{h,k+1})$ | Discriminator |
| $K$ | Option interval | $\mathcal{P}(\mathbf{s}_{h,k+1}|\mathbf{s}_{s,k},\mathbf{a}_{h,k})$ | Environment dynamics | $\pi_\theta(\mathbf{u}_h|\mathbf{s}_{h,0})$ | Scheduler |

**Output:** Learned parameters $\theta$, $\phi$, and $\psi$.

**Initialize:** Random model parameters $\theta$, $\phi$, and $\psi$; empty replay buffers $\mathcal{D}_{\text{scheduler}}$ and $\mathcal{D}_{\text{worker}}$.

**while** *termination not met* **do**

    /* Data collection                                                */

    **for** *scheduler step* $h = 0..\frac{T}{K} - 1$ **do**

        Sample an option $\mathbf{u}_h \sim \pi_\theta(\cdot|\mathbf{s}_{h,0})$.

        **for** *worker step* $k = 0..K - 1$ **do**

            Sample an action $\mathbf{a}_{h,k} \sim \pi_\phi(\cdot|\mathbf{s}_{h,k},\mathbf{u}_h)$.

            Step through the environment $\mathbf{s}_{h,k+1} \sim \mathcal{P}(\cdot|\mathbf{s}_{h,k},\mathbf{a}_{h,k})$.

            $\overline{\mathbf{a}}_{h,k}, \overline{\mathbf{s}}_{h,k+1} \leftarrow [\overline{\mathbf{a}}_{h,k-1}, \mathbf{a}_{h,k}], [\overline{\mathbf{s}}_{h,k}, \mathbf{s}_{h,k+1}]$

            $\mathcal{D}_{\text{worker}} \leftarrow \mathcal{D}_{\text{worker}} \cup (\mathbf{u}_h, \overline{\mathbf{a}}_{h,k}, \overline{\mathbf{s}}_{h,k+1})$

        **end**

        $R_h \leftarrow \sum_{k=0}^{K-1} r(\mathbf{s}_{h,k}, \mathbf{a}_{h,k}, \mathbf{s}_{h,k+1})$

        $\mathcal{D}_{\text{scheduler}} \leftarrow \mathcal{D}_{\text{scheduler}} \cup (\mathbf{s}_{h,0}, \mathbf{u}_h, \mathbf{s}_{h+1,0}, R_h)$

    **end**

    /* Model training                                              */

    **for** *batch* $m = 0..M - 1$ **do**

        /* Scheduler training                          */

        Uniformly sample transitions $\{(\mathbf{s}_t, \mathbf{u}_t, \mathbf{s}_{t+1})\}_{b=1}^B \sim \mathcal{D}_{\text{scheduler}}$.

        Compute gradient $\Delta\theta$ according to Eq. 3.

        Update models $\theta \leftarrow \theta + \alpha\Delta\theta$.

        /* Worker training                             */

        Uniformly sample transitions $\{(\mathbf{u}_h, \overline{\mathbf{a}}_{h,k}, \overline{\mathbf{s}}_{h,k+1})\}_{b=1}^B \sim \mathcal{D}_{\text{worker}}$.

        Compute intrinsic rewards $r_{h,k}^{lo} \leftarrow q_\psi(\mathbf{u}_h|\overline{\mathbf{a}}_{h,k}, \overline{\mathbf{s}}_{h,k+1})$.

        Compute gradient $\Delta\psi$ and $\Delta\phi$ according to Eq. 5.

        Update models $\phi \leftarrow \phi + \alpha\Delta\phi$ and $\psi \leftarrow \psi + \alpha\Delta\psi$.

    **end**

**end**

---

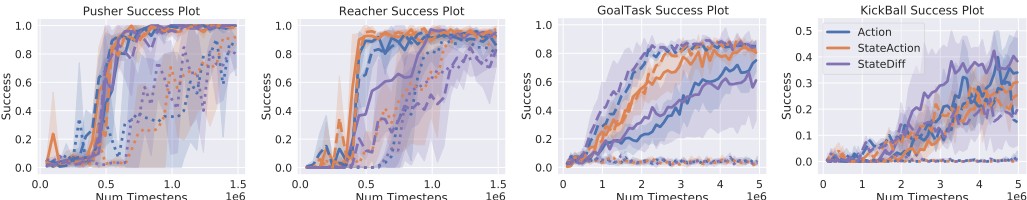

Figure 7: Comparisons of the mean success rates of three features of HIDIO (`Action`, `StateAction`, `StateDiff` at different option lengths $K$. Dotted lines indicate $K = 1$, solid lines indicate $K = 3$, and dashed lines indicate $K = 5$. $K = 3$ was used across all environments for the results in the main text.

## B   MORE ENVIRONMENT DETAILS

### B.0.1   PUSHER AND REACHER

These environments both have a time horizon of 100 with no early termination: each episode always runs for 100 steps regardless of goal achievement. For both, a success is when the agent achieves the goal at the *final* step of an episode. In REACHER, observations are 17-dimensional, including the positions, angles, and velocities of the robot arm, and in PUSHER observations also include the 3D object position. Both include the goal position in the observation space. Actions are 7-dimensional vectors for joint velocity control. The action range is $[-20, 20]$ in REACHER and $[-2, 2]$ in PUSHER.

There is an action penalty in both environments: at every timestep the squared $L_2$ norm of the agent action is subtracted from the reward. In PUSHER, this penalty is multiplied by a coefficient of $0.001$. In REACHER, it's multiplied by $0.0001$.

### B.0.2   GOALTASK AND KICKBALL

For both SOCIALROBOT environments, an episode terminates early when either a success is reached or the goal is out of range. For each episode, the positions of all objects (including the agent) are randomly picked. Observations are 18-dimensional. In GOALTASK, these observations include ego-centric positions, distances, and directions from the agent to different objects while in KICKBALL, they are absolute positions and directions. In KICKBALL, the agent receives a reward of 1 for successfully pushing a ball into the goal (episode termination) and 0 otherwise. At the beginning of each episode, the ball is spawned randomly inside the neighborhood of the agent. Three distractor objects are included on the ground to increase task difficulty. In GOALTASK, the number of distractor objects increases to 5. Both environments contain a small $L_2$ action penalty: at every time step the squared $L_2$ norm of the agent action, multiplied by $0.01$, is subtracted from the reward. GOALTASK has a time horizon of 100 steps, while KICKBALL's horizon is 200. Observations are 30-dimensional, including absolute poses and velocities of the goal, the ball, and the agent. Both GOALTASK and KICKBALL use the same navigation robot PIONEER2DX which has 2-dimensional actions that control the angular velocities (scaled to $[-1, 1]$) of the two wheels.

## C   OPTION DETAILS

### C.1   OPTION LENGTH ABLATION

We ablate the option length $K$ in all four environments on the three best HIDIO instantiations in Figure 7. $K = \{1, 3, 5\}$ timesteps per option are shown, with $K = 3$ and $K = 5$ performing similarly across all environments, but $K = 1$ performing very poorly in comparison. $K = 1$ provides no temporal abstraction, resulting in worse sample efficiency in PUSHER and REACHER, and failing to learn in GOALTASK and KICKBALL. Although $K = 5$ and $K = 3$ are generally similar, we see in GOALTASK that $K = 5$ results in better performance than $K = 3$ across all three instantiations, demonstrating the potential benefit of longer temporal abstraction lengths.

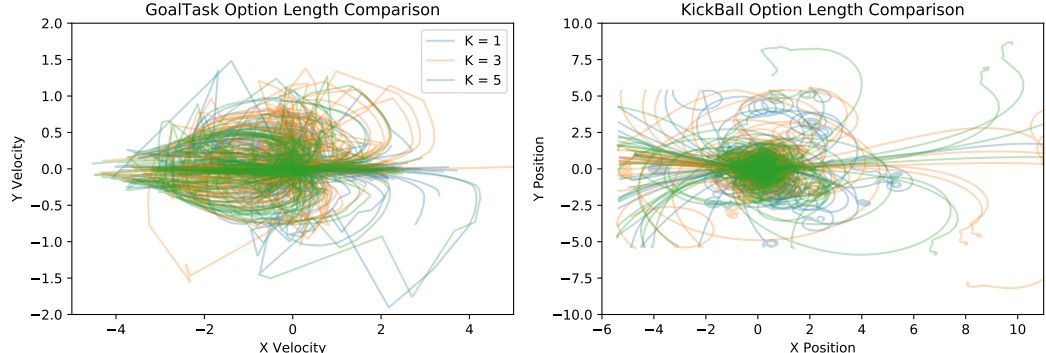

Figure 8: Trajectory distributions compared for different option lengths $K$ for the `StateAction` HIDIO instantiation in both SOCIALROBOT environments. These are obtained by randomly sampling an option uniformly in $[-1, 1]^D$ and keeping it fixed for the entire trajectory. 100 trajectories from each option are visualized and plotted in different colors.

We also plot the distribution of $(x, y)$ velocities[5] in GOALTASK and $(x, y)$ coordinates in KICK-BALL of randomly sampled options of different lengths in Figure 8. Despite the fact that these two dimensions only represent a small subspace of the entire (30-dimensional) state space, they still demonstrate a difference in option behavior at different option lengths. We can see that as the option length $K$ increases, the option behaviors become more consistent within a trajectory. Meanwhile regarding coverage, $K = 1$'s (blue) trajectory distribution in both environments is less concentrated near the center, while $K = 5$ (green) is the most concentrated at the center. $K = 3$ (orange) lies somewhere in between. We believe that this difference in behavior signifies a trade off between the coverage of the state space and how consistent the learned options can be depending on the option length. Given the same entropy coefficient ($\beta$ in Eq 5), with longer option lengths, it is likely that the discriminator can more easily discriminate the sub-trajectories created by these options, so that their coverage does not have to be as wide for the worker policy to obtain high intrinsic rewards. Meanwhile, with shorter option lengths, the shorter sub-trajectories have to be more distinct for the discriminator to be able to successfully differentiate between the options.

### C.2    OPTION VISUALIZATIONS

We visualize more option behaviors in Figure 9, produced in the same way as in Figure 5 and as detailed in Section 4.4. The top 4 picture reels are from KICKBALL. We see that KICKBALL options lead to varied directional driving behaviors that can be utilized for efficient navigation. For example, the second, third, and fourth highlight options that produce right turning behavior, however at different speeds and angles. The option in the third reel is a quick turn that results in the robot tumbling over into an unrecoverable state, but the options in the second and fourth reels turn more slowly and do not result in the robot flipping. The first option simply proceeds forward from the robot starting position, kicking the ball into the goal.

The bottom 4 reels are from PUSHER. Each option results in different sweeping behaviors with varied joint positioning and arm height. These sweeping and arm folding behaviors, when utilized in short sub-trajectories, are useful for controlling where and how to move the arm to push the puck into the goal.

### D    HYPERPARAMETERS

To ensure a fair comparison across all methods, we perform a hyperparameter search over the following values for each algorithm and suite of environments.

---

[5]Velocities are relative to the agent's yaw rotation. Because GOALTASK has egocentric inputs, the agent is not aware of the absolute $(x, y)$ coordinates in this task.

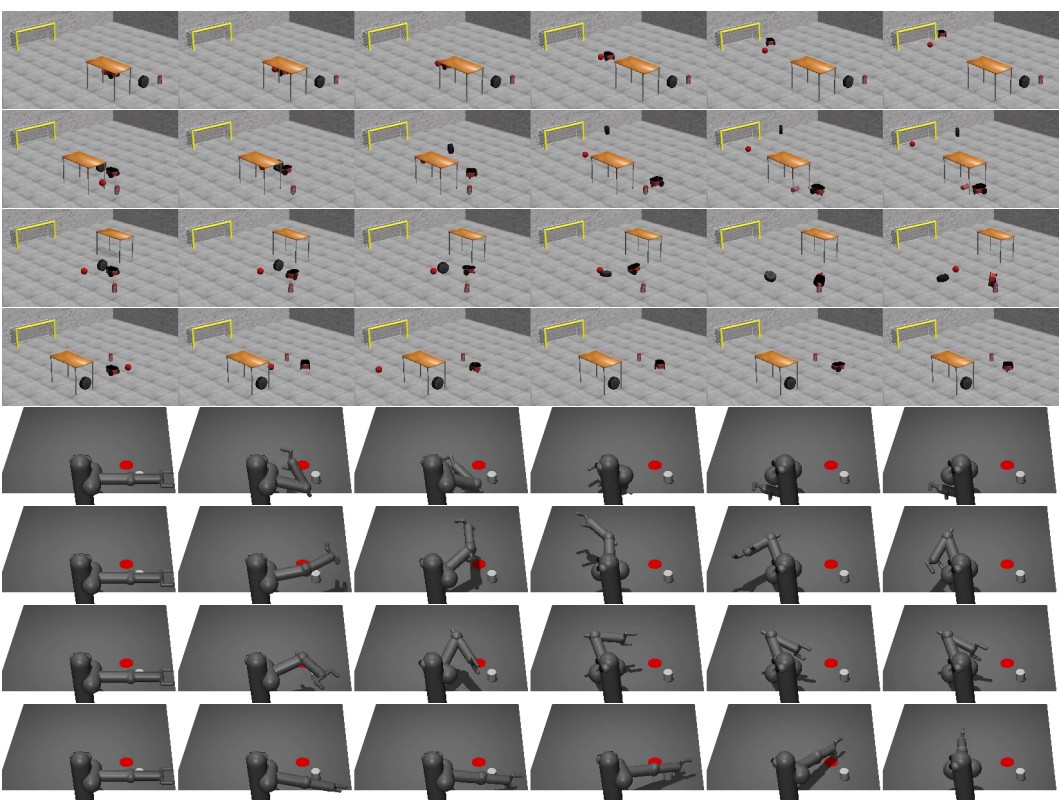

Figure 9: Eight example options from the `StateAction` instantiation on KICKBALL (top 4) and PUSHER (bottom 4).

### D.1 Pusher and Reacher

Shared hyperparameters across all methods are listed below (where applicable, and except when overridden by hyperparameters listed for each individual method). For all methods, we take the hyperparameters that perform best across 3 random seeds in terms of the area under the evaluation success curve (AUC) in the PUSHER environment.

- Number of parallel actors/environments per rollout: 20
- Steps per episode: 100
- Batch size: 2048
- Learning rate: $10^{-4}$ for all network modules
- Policy/Q network hidden layers: (256, 256, 256) with ReLU non-linearities
- Polyak averaging coefficient for target Q: 0.999
- Target Q update interval (training iterations): 1
- Training batches per iteration: 100
- Episodes per evaluation: 50
- Initial environment steps for data collection before training: 10000

Rollouts and training iterations are performed alternatively, one after the other. The *rollout length* searched below refers to how many time steps in each environment are taken per rollout/training iteration, effectively controlling the ratio of gradient steps to environment steps. A smaller roll-out length corresponds to a higher ratio. This ratio is also searched over for HIPPO and HIRO. Other hyperparameters searched separately for each algorithm are listed below, and selected ones are **bolded**.

#### D.1.1 SAC

- Target entropy min prob $\Delta$[6]: {0.1, **0.2**, 0.3}
- Replay buffer length per parallel actor: {50000, **200000**}
- Rollout Length: {12, 25, **50**, 100}

#### D.1.2 SAC w/ Action Repetition

- Action repetition length[7]: 3
- Rollout Length: {**4**, 8, 16, 33}

Other hyperparameters are kept the same as the optimal SAC ones.

#### D.1.3 HIDIO

The hyperparameters of HIDIO were mostly heuristically chosen due to the hyperparameter search space being too large.

- Latent option **u** vector dimension ($D$): {**8**, 12}
- Policy/Q network hidden layers for $\pi_\phi$ : (128, 128, 128)
- Steps per option ($K$): 3
- $\pi_\phi$ has a fixed entropy coefficient $\alpha$ of 0.01. Target entropy min prob $\Delta$ for $\pi_\theta$ is 0.2.
- Discriminator network hidden layers: (64, 64)
- Replay buffer length per parallel actor: {50000, **200000**}
- Rollout Length: {**25**, 50, 100}

---

[6]The target entropy used for automatically adjusting $\alpha$ is calculated as: $\sum_i [\ln(M_i - m_i) + \ln \Delta]$ where $M_i/m_i$ are the maximium/minimum value of action dim $i$. Intuitively, the target distribution concentrates on a segment of length $(M_i - m_i)\Delta$ with a constant probability.

[7]Chosen to match the option interval $K$ of HIDIO.

### D.1.4 HIRO

- Steps per option: {**3**, 5, 8}
- Replay buffer size (total): {500000, **2000000**}
- Meta action space (actions are relative, *e.g.*, meta-action is `current_obs + action`): `(-np.ones(obs_space - 3D_goal_pos)*2, np.ones(obs_space - 3D_goal_pos)*2)`
- Policy stddev noise: {**0.1**, 0.3, 0.5}
- Number of gradient updates per training iteration: {100, **200**, 400}

### D.1.5 HiPPO

For most hyperparameters, the search ranges chosen were derived after discussion with the first author of HiPPO.

- Learning rate: $3 \times 10^{-4}$
- Policy network hidden layers: (256, 256)
- Skill selection network hidden layers: {(32, 32), **(128, 64)**}
- Latent skill vector size: {5, **10**, 15}
- PPO clipping parameter: {0.05, **0.1**}
- Time commitment range: {(2, 5), **(3, 7)**}
- Policy training steps per epoch: {**25**, 50, 100}

### D.2 SOCIALROBOT

For all methods, we select the hyperparameters with the best area under the evaluation success curve (AUC) in the KICKBALL environment, and apply them to both KICKBALL and GOALTASK. The shared hyperparameters are as follows (if applicable to the algorithm, and except when overridden by the respective algorithm's list of hyperparameters):

- Number of parallel actors/environments per rollout: 10
- Steps per episode: 100 (GOALTASK), 200 (KICKBALL)
- Batch size: 1024
- Learning rate: $5 \times 10^{-4}$ for all network modules
- Policy/Q network hidden layers: (256, 256, 256) with ReLU non-linearities
- Polyak averaging coefficient for target Q: 0.95
- Target Q update interval (training iterations): 1
- Training batches per iteration: 100
- Episodes per evaluation: 100
- Evaluation interval (training iterations): 100
- Initial environment steps for data collection before training: 100000

The training terminology here generally follows section D.1.

### D.2.1 SAC

- Target entropy min prob $\Delta$: {0.1, **0.2**, 0.3}
- Replay buffer length per parallel actor: {20000, **100000**}
- Rollout length: {12, 25, **50**, 100}

### D.2.2 SAC W/ ACTION REPETITION

- Action repetition length[8]: 3
- Rollout Length: $\{4, 8, 16, \mathbf{33}\}$

Other hyperparameters are kept the same as the optimal SAC ones.

### D.2.3 HIDIO

Due to the large hyperparameter search space, we only search over the option vector size and rollout length, and select everything else heuristically.

- Latent option $\mathbf{u}$ vector dimension ($D$): $\{\mathbf{4}, 6\}$
- Policy/Q network hidden layers for $\pi_\phi$ (128, 128, 128)
- Steps per option ($K$): 3
- $\pi_\phi$ has a fixed entropy coefficient $\alpha$ of 0.01. Target entropy min prob $\Delta$ for $\pi_\theta$ is 0.2.
- Discriminator network hidden layers: (32, 32)
- Replay buffer length per parallel actor: 20000
- Rollout Length: $\{50, \mathbf{100}\}$

### D.2.4 HIRO

- Learning rate: $3 \times 10^{-4}$
- Steps per option: $\{\mathbf{3}, 5, 8\}$
- Replay buffer size (total): $\{\mathbf{500000}, 2000000\}$
- Meta action space (actions are relative, *e.g.*, meta-action is `current_obs + action`):
  - GOALTASK: `(-np.ones(obs_space) * 2, np.ones(obs_space) * 2)`
  - KICKBALL: `(-np.ones(obs_space - goal_space) * 2, np.ones(obs_space - goal_space) * 2)` (because the goal position is given but will not change in the observation space)
- Policy stddev noise $\{\mathbf{0.1}, 0.3, 0.5\}$
- Number of gradient updates per training iteration: $\{100, 200, \mathbf{400}\}$

### D.2.5 HIPPO

- Learning rate: $3 \times 10^{-4}$
- Policy network hidden layers: $\{(\mathbf{64}, \mathbf{64}), (256, 256)\}$
- Skill selection network hidden layers: $\{(\mathbf{32}, \mathbf{32}), (128, 64)\}$
- Latent skill vector size: $\{4, \mathbf{8}\}$
- PPO clipping parameter: $\{\mathbf{0.05}, 0.1\}$
- Time commitment range: $\{(\mathbf{2}, \mathbf{5}), (3, 7)\}$
- Policy training steps per epoch: $\{25, \mathbf{50}, 100\}$

---

[8]Chosen to match the option interval $K$ of HIDIO.

