# OpenReview forum: "Hierarchical Reinforcement Learning by Discovering Intrinsic Options"
_ICLR.cc/2021/Conference — ICLR 2021 Poster_

### Official Review · AnonReviewer4 · 2020-10-28

**Rating:** 4
**Confidence:** 3

**Review:**

The paper describes an interesting approach to self-supervised option learning for hierarchical reinforcement learning. In particular, the authors propose to learn options through an intrinsic entropy minimization objective conditioned on option trajectories. Empirical studies are performed to demonstrate the success rate and sample efficiency of this method over two state-of-the-art methods.

The key concern that I have with this paper is its organization and presentation. I am concerned that the paper takes too much space describing what other methods do not do (throughout the contributions sections), rather than concretely describing the approach. Furthermore, many sections seem to be an aggregation of contributions, related work, and results, which further makes the paper hard to follow. I believe that this does the paper a disservice by decreasing readability. Therefore, I am currently voting for rejecting the paper.

Questions and Comments:
1) In the Introduction, the authors claim that their approach requires "little" manual design? Can this claim be better grounded? What design is required?
2) There is extensive work on option discovery [1,2]. as well as learning decompositions for reinforcement learning [3-6]. I would have liked to see a more extensive literature review comparing and contrasting with these works.
3) On page 2, the authors say, "for the purpose of this paper, we consider a hierarchy of two levels". However, it is not clear to me how this approach would extend to a hierarchy with more than two levels. Can the authors please clarify this?
4) What is k? h? (page 2)
5) On page 3, the authors claim that, unlike prior work, their formulation has no constraints on what the options can be. Can the authors comment on this claim in more detail?
6) In section 3.2, the authors mention the use of goals. Are the authors considering goal-conditioned MDPs [7]? If so, this should be made more explicit.
8) The MDP definition in section 3.2 is unclear. Could the authors please clarify this definition?
9) What is meant by "resolution" (section 3.2)?
10) Why does the objective defined by the authors enable skills that have more diverse semantics than previous approaches (claimed on page 4, section 3.2)?
11) Page 5, section 3.4: is the importance correction not used at all (for either the scheduler or discriminator)?
12) What assumptions are needed for this method to work well?
13) In the experiment section, it says that in the pusher environment, success means that the goal is achieved in the final step of the episode. Does this mean that pushing the object to the position does not terminate the episode and, instead, the object must remain there until the episode terminates in a predetermined number of steps?
14) Can the authors please clarify the purpose of the additional suite of environments? What do these environments enable that existing ones do not?

[1] Brunskill, et al., PAC-inspired option discovery in lifelong reinforcement learning. ICML, 2014.
[2] Topin, et al., Portable option discovery for automated learning transfer in object-oriented Markov decision processes. IJCAI, 2015.
[3] Mehta et al., Automatic discovery and transfer of MAXQ hierarchies. ICML, 2008.
[4] Winder et al., Planning with abstract learned models while learning transferable subtasks. AAAI, 2020.
[5] Li, et al., An efficient approach to model-based hierarchical reinforcement learning. AAAI, 2017.
[6] Rafati, et al., Learning representations in model-free hierarchical reinforcement learning. AAAI, 2019.
[7] Nasiriany, et al., Planning with goal-conditioned policies. NeurIPS, 2019.

---

> ### Author Response · Authors · 2020-11-13
> **Author Response to AnonReviewer4 (3/3)**
>
> > “9. Why does the objective defined by the authors enable skills that have more diverse semantics than previous approaches (claimed on page 4, section 3.2)?”
>
> As already explained in the same paragraph, the previous approaches [Gregor et al., 2016] and [Eysenbach et al., 2019] always define an option (skill) with respect to the initial state $s_0$ sampled from a fixed initial distribution. The semantics of an option spans over the entire episode starting from $s_0$ in their case. Our options are defined with respect to $s_0, s_K, s_{2K}, \cdots$ because we are using sub-trajectories of length $K$ to identify options. So in our case, the same option vector (e.g., [1.0, 1.0, 1.0]) could mean different things given different values of $s_K$, depending on which state the agent is in currently in the middle of an episode.
>
> > “10. Page 5, section 3.4: is the importance correction not used at all (for either the scheduler or discriminator)?”
>
> Correct. We tried including the importance correction but it yields high variance in importance ratios and actually hinders the training. Similar observations were made by [Nachum et al. 2018; Fedus et al. 2020]. A baseline without importance correction was tested in [Nachum et al. 2018] and had competitive performance compared to their full method. We also found that our method is able to perform well empirically even without importance correction. We include that paragraph for discussions and inform readers what we’ve tried but didn’t work in practice.
>
> > “11. What assumptions are needed for this method to work well?”
>
> One important assumption is that by optimizing the worker objective Eq (5), the discovered options can actually be assembled by the scheduler to accomplish the environment task. This assumption can be made true if 1) the discovered options are diverse (differentiable from each other) and 2) the population of the discovered options covers the trajectory space well (made possible by maximizing the entropy of the worker policy as in SAC [Haarnoja et al., 2018]).
>
> > “12. In the experiment section, it says that in the pusher environment, success means that the goal is achieved in the final step of the episode. Does this mean that pushing the object to the position does not terminate the episode and, instead, the object must remain there until the episode terminates in a predetermined number of steps?”
>
> Correct. The pusher environment doesn’t have an early termination. Each episode will run for a fixed number of steps. We’ve clarified this in the revision. This setup is more challenging as it can test if the agent actually knows how to achieve the task or simply accomplishes it accidentally when exploring.
>
> > “13. Can the authors please clarify the purpose of the additional suite of environments? What do these environments enable that existing ones do not?”
>
> Many environments used in other hierarchical works are navigation and goal-reaching tasks that have *dense* rewards (such as some of the Ant tasks in HIRO [Nachum et al. 2018]) or are different variants of the same navigation task (such as the Snake/Ant Gather tasks in HIRO and HiPPO [Li et al. 2019]). Meanwhile, all of our environments are sparse-reward, continuous action-space tasks that serve different purposes and are based upon real robots. While our reaching task can be thought of as a sparse-reward navigation task, our pushing task is a robotic object manipulation task, and both simulate a one-armed PR2 robot (https://blog.robotiq.com/bid/65419/Collaborative-Robot-Series-PR2-from-Willow-Garage). Furthermore, GoalTask is a more complex navigation task that involves randomly placed distractor objects that the agent must learn to avoid and not get stuck on, and Kickball is difficult in that the agent must learn to perform the skills required to solve GoalTask and then also learn to kick a ball into a goal, all under sparse reward supervision. These two tasks emulate the Pioneer wheeled robot (https://www.generationrobots.com/media/Pioneer3DX-P3DX-RevA.pdf).

---

> ### Author Response · Authors · 2020-11-13
> **Author Response to AnonReviewer4 (2/3)**
>
> > “3. On page 2, the authors say, "for the purpose of this paper, we consider a hierarchy of two levels". However, it is not clear to me how this approach would extend to a hierarchy with more than two levels. Can the authors please clarify this?”
>
> Most existing hierarchical methods consider only two levels, one exception we can find is [Levy et al., ICLR 2019] which considers three levels with non-trivial assumptions and engineering efforts. However, a potential way of extending our approach to multiple levels is as follows. Suppose the total number of levels is $N$. For the higher $N-1$ levels, each level learns a policy that conditions on the option output by the level above (the highest level doesn’t condition on any option), and outputs a “finer-grained” option to the level below. The lowest level ($N$) outputs the actual action to interact with the environment. Each of the higher $N-1$ levels uses the same form of intrinsic option objective Eq (5) defined on subtrajectores at different time scales: the highest level uses the most abstract (downsampled) sub-trajectory to discover options, and the abstraction decreases as the level descends. Suppose the policy at each level has a time budget of $K$ under an option from the level above, then the abstraction at the level i is roughly $(1/K)^{N-i}$. In general, this is feasible for on-policy training because policies at all levels are tuned together. For off-policy training, effective techniques that address the non-stationary issue (lower-level policies keep changing and confuse the higher levels) might be needed, which is still an open problem. We leave the multiple-level hierarchy to our future work.
>
> > “4. What is k? h? (page 2)”
>
> As defined in Eq (2), $h$ is the time step index of the higher-level scheduler, and $k$ is the time step index of the lower-level worker. A horizon of length $T$ is divided into $H$ options each indexed by $h$, where within each option window the time steps are indexed by $k$.
>
> > “5. On page 3, the authors claim that, unlike prior work, their formulation has no constraints on what the options can be. Can the authors comment on this claim in more detail?”
>
> The final learned options serve as latent variables to modulate the worker policy as inputs. Some existing methods like HIRO [Nachum et al., NeurIPS 2018] specifically require that the goal space is a subspace of the state space, so that the addition operation between a goal and a state is meaningful in their task. The downside is that if the state space is high-dimensional (e.g., images), then this is no longer applicable.
>
> > “6. In section 3.2, the authors mention the use of goals. Are the authors considering goal-conditioned MDPs [7]? If so, this should be made more explicit.”
>
> Our hierarchical framework is more closely related to the options framework [Sutton et al., 1999]. The use of the word “goal” is just for exposition and a comparison to goal-conditioned RL. Informally we can think of an option similar as a goal when modulating the behavior of a policy (used as its input). To avoid confusions, we’ve unified the terms “goal”, “option”, and “skill” as “option” in the revised text as suggested by AnonReviewer3.
>
> > “7. The MDP definition in section 3.2 is unclear. Could the authors please clarify this definition?”
>
> This new MDP equips the worker with historical state and action information since the time $(h,0)$ when an option $h$ was scheduled. More specifically, we define the new MDP so that now each (meta) state for the lower-level worker policy consists of a sub-trajectory of the states of the original MDP. These are defined so that in every option window $h$ of length $K$, the worker policy and discriminator will see the history starting from the beginning step $(h,0)$ of the option up to the current step $(h,k)$. We’ve added this further interpretation in section 3.2.1.
>
> > “8. What is meant by "resolution" (section 3.2)?”
>
> We use the word “resolution” to informally describe how fine-grained a discriminator model could divide a continuous feature space. A powerful discriminator model (high capacity) could successfully classify two vectors (sub-trajectories in our case) into totally different classes (options in our case) even though these sub-trajectories only slightly differ. For example, consider two vectors [1.0, 1.0, 1.01] and [1.0, 1.0, 1.0]. If the discriminator is powerful enough, it’s possible that it learns to classify them into different labels, in which case we call high resolution informally. This causes problems because even though the discriminator thinks the discovered options are diverse, they actually only cover a small portion of the sub-trajectory space. As a result, the scheduler policy might find it difficult to utilize these options as they are too similar for the environment task. To resolve this issue, “we add a second term which computes the entropy of the worker policy” to Eq (4).

---

> ### Author Response · Authors · 2020-11-13
> **Author Response to AnonReviewer4 (1/3)**
>
> Thank you for your detailed review and feedback.
>
> > “I am concerned that the paper takes too much space describing what other methods do not do (throughout the contributions sections), rather than concretely describing the approach.”
>
> We assume that by “contributions sections”, the reviewer refers to the introduction section 1. We’ve updated section 1 to include more concrete descriptions about the novel part of our approach.
>
> > “Furthermore, many sections seem to be an aggregation of contributions, related work, and results, which further makes the paper hard to follow.”
>
> Our paper has 5 sections (excluding the conclusion section).
>  1. It is a common practice to include contributions, related work, and some rough results in the first introduction section to give the reader a general picture.
>   2. Section 2 is for background and preliminaries; it does not contain contributions, related work, or results.
>   3. The entire section 3 is devoted to describing our approach. There might be some confusions here because in several places, before/after we present one formula or technical detail, we tend to use the current context to compare our technique to existing ones for better calibration, making readers better informed (e.g., after Eq (3), after Eq (5), before/after we instantiate the discriminator, etc). We’ve updated the first paragraph of section 3 and the first paragraph of section 3.2 for less comparison but more emphasis on what we do. Note that section 3 does not contain any experimental results. Specifically, section 3.3 is still about the approach itself (how we instantiate the discriminator); it doesn’t show any experiment results.
>   4. We believe that we only present our results in the experiment section 4. It’s fully devoted to results, without talking about contributions and related work.
>   5. Lastly, most of the related work discussion is presented in section 5. It’s usual to mention the contributions again in this section. Also this section doesn’t contain any results.
>
> It would be better if the reviewer can give some specific examples of an aggregation of contributions, related work, and results, so that we will be able to better address the reviewer’s readability concern.
>
> > “I believe that this does the paper a disservice by decreasing readability. Therefore, I am currently voting for rejecting the paper.”
>
> Both AnonReviewer3 and AnonReviewer5 agree that the paper is written well (“This paper is in general well-written with clarity”; “The paper is generally organised well and written clearly”). We do believe that this readability concern has been largely addressed by our responses and the revised manuscript.
>
> > “1. In the Introduction, the authors claim that their approach requires "little" manual design? Can this claim be better grounded? What design is required?”
>
> The context here is that our hierarchical method is being compared to existing hRL methods, regarding the assumption or design about the task structures and option space. Many existing hRL methods have to design the task structure (e.g., the task of fetching an object is designed to have three stages: first reaching the object, then picking up the object, and finally coming back to the initial position. Note that this decomposition is not inherent to the task but designed to enable lower-level subtasks), or the skills needed to solve the task (e.g., pre-programmed motor skills), or the structure of the option space (e.g., given an option vector, the first N dims are the target xyz location and the remaining M dims are the target velocities). Our method generally doesn’t assume such prior knowledge since it’s hard to generalize across tasks, and our options are task agnostic and are learned from scratch. On the other hand, we still have general algorithmic designs such as hyperparameter and network selection.  To avoid confusions, we’ve updated the manuscript to clarify this and state that our approach requires less manual design in the task domain.
>
> > “2. There is extensive work on option discovery [1,2]. as well as learning decompositions for reinforcement learning [3-6]. I would have liked to see a more extensive literature review comparing and contrasting with these works.”
>
> Thank you for these suggestions, we have incorporated them into the related works section of our updated manuscript.

---

### Official Review · AnonReviewer1 · 2020-10-28
**Review of Hierarchical Reinforcement Learning by Discovering Intrinsic Options: little to say about options**

**Rating:** 4
**Confidence:** 2

**Review:**

#######################################################################

Summary:

In this paper the authors present a new method for hierarchical reinforcement learning, demonstrated with a 2 layer architecture, in which the higher layer Scheduler policy choose lower level Worker policies at fixed intervals. The lower level Worker policies (options) are trained through an intrinsic entropy minimization.

#######################################################################

Reasons for score:

This paper seems to be based strongly on the work in Sharma et al., 2019b and Eysenbach et al., 2019, and while it makes a number of interesting modifications to those methods there is in sufficient decoupling / ablation of the issues. I think the paper would be notably stronger with additional experimental evidence

#######################################################################Pros:

1. The paper makes some reasonably sensible extension to well-know methods.

2. The authors treatment of hyper parameters and implementation details looks to be thorough

#######################################################################

Cons:

1. I found parts of the paper difficult to parse. For example it was not immediately clear to me what exactly an option $\bf{u} \in [-1,1]^D$ is... What values can / does D take? Another free parameter? Is $\bf{u}$ a latent variable that yields an option through the worker policy? (this terminology would seem to better align with that in Eysenbach et al., 2019 and the broader literature around options). I'm also not sure $\bar{\bf{s}}_t$ (single subscript) is defined?

2. As the authors note the paper introduces a number of new elements: the reset of the worker policy every k-step, the utilization of full sub-trajectories, and the various instantiations of the discriminator... but these are never unwound. It would seem to be the case that there would be a fair bit of interplay between these elements. Additionally it appears that there is some need to modify the implementation to support these choices - most notably the discounting for the "shortsighted worker"... is this still necessary if full sub-trajectories are not used?

3. The experimentation did not address some fairly obvious enquires (what is the effect of different choices of K in your experiments? How would a practitioner approach choosing this value?)

4. There is remarkably little in this paper that talks about the actual options uncovered via this method. Does the method uncover different options for different values K? What do they look like? etc.

#######################################################################

Questions during rebuttal period:

Q1: a fair amount of the focus of the paper appears to be on the instantiation of the discriminator - do you consider that to be a central contribution?

Q2: you talk about the fact that a key divergence from priori work is that there is "no constraint on the value on what $\bf{u}$ could represent." Can you talk about why this is a good thing and have you explored when it might not be such a good thing?

#######################################################################

---

> ### Author Response · Authors · 2020-11-13
> **Author Response to AnonReviewer1 (1/3)**
>
> Thank you for your review and detailed feedback.
>
> > This paper seems to be based strongly on the work in Sharma et al., 2019b and Eysenbach et al., 2019
>
> Please see our response to your Q1 below for a summary of major differences with their methods, and a clarification of our central contributions.
>
> > and while it makes a number of interesting modifications to those methods there is insufficient decoupling / ablation of the issues.
>
> The reviewer might think that the proposed elements (e.g., use of sub-trajectories, shortsighted worker, discriminator instantiation) are exclusive or completely independent of each other, however, this is not true. Please see our detailed response to your question “As the authors note the paper introduces a number of new elements…” below. In a word, these new elements are essential components of realizing the single worker objective in Eq (5). They are definitely not unrelated improvements proposed from different angles.
>
> Besides, we believe that we did provide enough ablation studies for our approach:
>
> 1. In section 4.1, we explored 6 different instantiations of the discriminator and found that roughly 3 instantiations (Action, StateAction, and StateDiff) can perform reasonably well (Figure 3). This disentangles the instantiation choice with the other components of the framework.
> 2. We also compare the SOFT and HARD versions of the worker policy in section 4.1. Note that their implementation difference is only a hyperparameter (discounting factor). So they are not two different approaches but ablations of our approach.
> 3. In section 4.3, we explore another ablation baseline that pretrains the worker policy, to explore if it’s necessary for a joint training of the worker and scheduler.
>
> > “I think the paper would be notably stronger with additional experimental evidence”
>
> ~~We are running a small experiment and obtaining some extra results for different values of $K$ (the option length)~~ We have added results for ablations with different option lengths (see edits in the second reply to this review). Given the existing ablation studies we have performed (summarized above), it would be better if the reviewer can further suggest what additional experiments are needed.
>
> > “1. I found parts of the paper difficult to parse. For example it was not immediately clear to me what exactly an option $u \in [-1, 1]^D$ is... What values can / does D take? Another free parameter? Is $u$ a latent variable that yields an option through the worker policy? (this terminology would seem to better align with that in Eysenbach et al., 2019 and the broader literature around options). I'm also not sure $\bar{s}$ (single subscript) is defined?”
>
> An option $u$ can be any valid point in the defined space $[-1, 1]^D$, and yes it can be treated as a latent variable to modulate the worker policy. The value of $D$ is a hyperparameter (see appendix D.1.3 and D.2.3 for the values we chose in the experiments). We’ve clarified this in the revision (section 3, paragraph 2). For the single subscript t, we intended to use it to refer to a general time step $0 \le t < T$, and any $t = h * K + k$ for a pair of $(h, k)$ (see the line right before Eq (2)). But to avoid the confusion, we’ve removed the use of single subscripts $t$ completely in the revision.

---

> ### Author Response · Authors · 2020-11-13
> **Author Response to AnonReviewer1 (2/3)**
>
> > “2. As the authors note the paper introduces a number of new elements: the reset of the worker policy every k-step, the utilization of full sub-trajectories, and the various instantiations of the discriminator... but these are never unwound. It would seem to be the case that there would be a fair bit of interplay between these elements. Additionally it appears that there is some need to modify the implementation to support these choices - most notably the discounting for the "shortsighted worker"... is this still necessary if full sub-trajectories are not used?”
>
> We are not quite sure what the reviewer means by “these are never unwound”. In fact, all these elements together realize the single worker objective in Eq (5). They are *not* exclusive and each of them can’t function in a standalone way.
>
> We don’t need to make any implementation modification to support these choices. Note that we always have a *single unified* implementation that contains all these elements mentioned, and they together constitute our approach. The implementation of all the content of HIDIO has only roughly 1.1k lines of Python code in total, based on a well established open-source RL framework. We will open source our implementation once the manuscript is accepted.
>
> From the reviewer’s comments, we believe that there might be some misunderstanding regarding the use of sub-trajectories, how the worker policy does inference, and what the objective in Eq (5) indicates. We’ve provided a detailed explanation in the response to Q1 “a fair amount of the focus …”. To answer the last sentence above, note that even though the worker policy is modulated by options (instead of being manually “reset”) that change every K-th step, the decisions made in the current option window of length $K$ *will* affect the experiences in the next option window, due to the fact that the last state $s_{h,K}$ in the current option window is defined to be the first state $s_{h+1,0}$ in the next option window. **Crucially, the worker policy is trained not only to discover options at the current option window, but also to make the discovery in future easier**. By decreasing the discount factor for the worker policy, we are essentially weakening this temporal relation to make it more “shortsighted” and focused on optimizing the present. This only requires changing a single hyperparameter: the discount factor for the worker policy. We are not sure about how we will be able to not use full sub-trajectories, as suggested by the reviewer. If the reviewer can further clarify the question in the last sentence, we will provide more explanations accordingly.
>
> > “3. The experimentation did not address some fairly obvious enquires (what is the effect of different choices of K in your experiments? How would a practitioner approach choosing this value?)”
>
> We empirically set $K$ to 3. Ideally we should run a grid search over several values (e.g., $K=1, 3, 5, 9, 13, \ldots$) to determine the best one just like what we did for other baselines in the appendix. But given the limited time budget then, we only searched over the option vector size ($D$) and rollout length, and selected everything else heuristically. From our experience, if $K$ is too small, then the induced little temporal abstraction won't bring much benefit to the higher-level scheduler policy; if $K$ is too large compared to the total episode length, then the agent might have difficulty mastering fine-grained skills because options are self-supervised learned. ~~We’ve started a new extra experiment of different $K$ values, and will include the results once the running experiment finishes.~~ We have added Figures 7 and analysis in appendix Section C.1 which compares performance across all tasks with different option lengths.
>
> > “4. There is remarkably little in this paper that talks about the actual options uncovered via this method. Does the method uncover different options for different values K? What do they look like? etc.”
>
> In section 4.4 and Figure 5, we showed two examples of the discovered options. We have now added Appendix Section C.2 and Figure 9, which contain analysis and visualizations of 8 more options,  and ~~we will add a comparison of different options by different values of $K$ after the running experiment finishes.~~ we have added Figure 8 and analysis in appendix Section C.1 which visualizes the distribution of trajectories with different option lengths in GoalTask and KickBall.

---

> ### Author Response · Authors · 2020-11-13
> **Author Response to AnonReviewer1 (3/3)**
>
> > “Questions during rebuttal period: Q1: a fair amount of the focus of the paper appears to be on the instantiation of the discriminator - do you consider that to be a central contribution?”
>
> The instantiation of the discriminator only takes section 3.3 (half a page), and it’s less than half of the worker policy section 3.2. Both these two sections are central contributions of our paper. To clarify, note that section 3.2 is not just a simple review of previous methods, in fact it presents a new formulation in Eq (5) given that:
> 1. We constructed a new meta MDP on top of the original MDP to enable predicting options based on sub-trajectories. Without this, one simply can’t learn a worker policy from a reward function that is defined by *multiple* time steps (sub-trajectories) because it’s no longer Markovian (e.g., the reward at step t is defined by $K$ steps behind). This new meta MDP is critical and we’ve further addressed its importance in section 3.2.1. It also greatly affects the input space of the worker policy (stacking $K$ input frames).
> 2. More importantly, Eq (5) implies that actions taken by the worker policy under the current option will have consequences on the next option. This is because the final state $s_{h,K}$ of the current option is defined to be the initial state $s_{h+1,0}$ of the next option. So in general, the worker policy is trained not only to discover diverse options across the current $K$ steps, but also to make the discovery easier in the future steps. Furthermore, the optimization of the worker at the current option *is* in fact affected by the scheduler due to newly scheduled options in the future (expectation is taken over the scheduler policy). In other words, the worker policy needs to solve the credit assignment problem *across* options, under the expectation of the scheduler policy. These properties are missing in either DIAYN [Eysenbach et al., 2019] or VIC [Gregor et al., 2016] since they always consider full-episode option discovery in a non-hierarchical setting (i.e., one level). We’ve added a paragraph to address this difference in the revision (section 3.2.1).
>
> Given the above two important aspects of Eq (5), we believe that section 3.2 (worker policy) is also a central contribution of our paper.
>
> > “Q2: you talk about the fact that a key divergence from priori work is that there is "no constraint on the value on what $u$ could represent." Can you talk about why this is a good thing and have you explored when it might not be such a good thing?”
>
> By “no constraint on what $u$ could represent”, we mainly refer to the fact that the space of $u$ doesn’t have predefined semantics that puts constraints on its values or structure. A counterexample would be HIRO [Nachum et al., NeurIPS 2018] which specifically requires that the goal (similar to our option in our case) space is a subspace of the state space and an addition operation can be performed between a goal vector and a state vector in order to manage the lower-level policy. This kind of assumption or constraint requires prior knowledge of the task at hand, and it might be difficult to generalize to other tasks (e.g., where the state space is a high-dimensional image space). Thus a redesign or adaptation of the method is needed for a new task. On the contrary, our option $u$ is task-agnostic and simply a latent variable. The benefit of this is that it’s totally learned by the method itself from scratch given the downstream task, and can be easily retrained for a new task. Of course, an issue is that it can take more effort to debug or interpret the learned representation of $u$, and some visualization techniques might be needed for that.

---

### Official Review · AnonReviewer5 · 2020-11-06
**a novel hRL method that learns options in a self-supervised way**

**Rating:** 7
**Confidence:** 3

**Review:**

This paper present HIDIO, a hierarchical RL method that leverages self-supervised losses to discover intrinsic options while learning a scheduler to leverage the learned options to optimize the accumulated reward. HIDIO differentiates from prior work through enabling the low-level network/worker to discover task-agnostic options that can be generalized to future tasks, thus requiring no pre-training of skills,  and at the same time makes minimal assumption about the task structure.

This paper is in general well-written with clarity. The proposed  method is technically sound and empirically outperforms existing methods. The experiments in four different tasks first demonstrate an ablation of different feature extractors used for the self-supervised loss and analyzed the each of their advantages/disadvantages; followed by experiments comparing HIDIO with existing hRL methods and show that HIDIO outperforms in all tasks. Although the experiments are conducted in two different domains and four different tasks, all four tasks seem to be similar in nature (i.e. all pushing and reaching tasks). It would be great to see how HIDIO compare with other methods in more complex task domains such as in the work of [1] and [2], where the skill discovery network need to work with high dimensional raw input data and computing state-similarity can be tricky.


[1] Lynch, C., Khansari, M., Xiao, T., Kumar, V., Tompson, J., Levine, S., & Sermanet, P. (2020, May). Learning latent plans from play. In Conference on Robot Learning (pp. 1113-1132).
[2]Chuck, C., Chockchowwat, S., & Niekum, S. (2020). Hypothesis-Driven Skill Discovery for Hierarchical Deep Reinforcement Learning. International Conference on Intelligent Robots and Systems, 2020.

---

> ### Author Response · Authors · 2020-11-13
> **Author Response to AnonReviewer5**
>
> Thanks for the positive review and paper feedback.
>
> ***
> > Although the experiments are conducted in two different domains and four different tasks, all four tasks seem to be similar in nature (i.e. all pushing and reaching tasks).
>
> Regarding the similarity of the four different tasks, we believe that while they are somewhat similar in nature, they are difficult in different ways and test the ability of agents to learn options not present in the commonly used navigation tasks in other hierarchical RL works. For a detailed analysis of what each task is testing and how they’re different, please see our response to AnonReviewer4’s Question 13 (“Many environments used in other hierarchical works are navigation…”).
> ***
>
> > It would be great to see how HIDIO compare with other methods in more complex task domains such as in the work of [1] and [2], where the skill discovery network need to work with high dimensional raw input data and computing state-similarity can be tricky.
>
> The complex tasks in the works you referenced are very interesting, although we are currently focusing on state-based rather than image-based environments. In theory, we believe that HIDIO can be readily applied to image-based environments with CNNs instead of MLPs as the early layers of the policy networks and discriminator, without changing the overall framework and objectives. However, one might need to resolve some practical issues induced during this process (mostly hyperparameter and network selection), and we leave this extension to future work. We’ve added the mentioned works to our related works section (last part of section 5).
> ***

---

### Official Review · AnonReviewer3 · 2020-11-10
**The paper develops a new hierarchical reinforcement learning algorithm that yields good results in four robotic manipulation and navigation tasks. The analysis is well done. The paper is clear and well structured.**

**Rating:** 8
**Confidence:** 4

**Review:**

The paper develops a hierarchical reinforcement learning algorithm and analyzes its behaviour in four robotic manipulation and navigation tasks. The approach is based on a two-level hierarchy, *scheduler* at the top and *worker* at the bottom. This is similar to other approaches in the literature and the algorithm uses many ideas and elements from existing algorithms. However, these ideas and elements are combined in a novel and well-justified manner. The result is an algorithm that yields good results in a range of problems. The experiments are well done. The paper is generally organised well and written clearly. Relevant literature is reviewed well.

The paper can be improved by a more comprehensive and detailed analysis of the behaviour of the algorithm, in particular, the options that are found. Section 4.4 is useful but very short. It describes only two options. Perhaps such an analysis can be added to the appendix.

In the proposed algorithm, the scheduler outputs an option every K steps in the environment. It would be reasonable to question whether more flexibility would be useful here, one that allows for varying option durations.

The authors state in Section 2 that they 'will use the terms “goal”, “option”, and “skill” interchangeably.' In the literature, these terms refer to related but different concepts. Using them interchangeably is not good scientific practice.

I did not find Figure 1 particularly useful. The simple and intuitive structure of the algorithm does not come through in the figure.

---

> ### Author Response · Authors · 2020-11-13
> **Author Response to AnonReviewer3**
>
> Thank you for your positive review and feedback.
> ***
> > The paper can be improved by a more comprehensive and detailed analysis of the behaviour of the algorithm, in particular, the options that are found. Section 4.4 is useful but very short. It describes only two options. Perhaps such an analysis can be added to the appendix.
>
> Per your recommendation, we have now added Appendix Section C, which contains analysis of 8 more options in two environments, and Figure 9, which visualizes those 8 options. ~~Furthermore, we are running an extra experiment by ablating $K$ across all four environments, and we will add extra figures and analyses regarding option behaviors at different $K$ values once this finishes. ~~ Furthermore, we have added Figures 7 and 8 and analysis in appendix Section C.1 which compares performance across all tasks with different option lengths and visualizes their distribution of trajectories in GoalTask and KickBall.
> ***
> > In the proposed algorithm, the scheduler outputs an option every K steps in the environment. It would be reasonable to question whether more flexibility would be useful here, one that allows for varying option durations.
>
> Using a fixed value of $K$ is a simplification in our method since our focus in this paper is Eq (5) and the various instantiations of the discriminator. Many existing methods like HIRO [Nachum et al., NeurIPS 2018] and HiPPO [Li et al., 2020] also assume a fixed $K$. It’s indeed interesting to question if varying option durations would benefit the method or not. However, that involves a non-trivial addition to the method (e.g., learning another network head in the scheduler policy to decide the option duration). Such a question can be considered for future work.
> ***
> > The authors state in Section 2 that they 'will use the terms “goal”, “option”, and “skill” interchangeably.' In the literature, these terms refer to related but different concepts. Using them interchangeably is not good scientific practice.
>
> Thanks for pointing out this issue. Originally we used them interchangeably for an easier comparison with other similar methods from different backgrounds. We realize that it’s indeed confusing in the current way and have changed to use the word “option” throughout the paper.
> ***
> > I did not find Figure 1 particularly useful. The simple and intuitive structure of the algorithm does not come through in the figure.
>
> Figure 1 has been replaced by a more straightforward illustration that focuses on the hierarchical interplay between the scheduler and worker policy.
> ***

---

### Author Response · Authors · 2020-11-13
**Summary of Responses and Paper Revisions**

We thank all the reviewers for their constructive comments. Both AnonReviewer3 and AnonReviewer5 think that the method is novel and technically sound, the experiment results are empirically strong, and the paper is generally well written. While AnonReviewer1 believes that our work “makes some reasonably sensible extensions to well-known methods”, the main concern raised by AnonReviewer1 is “insufficient decoupling and ablation”. However, we believe that this is probably due to misunderstanding the overall HIDIO framework (e.g., the proposed new elements are in fact nonexclusive and compatible) and our central contributions, and overlooking the existing ablation studies in our experiments. The main concern raised by AnonReviewer4 is the readability issue, even though both AnonReviewer3 and AnonReviewer5 acknowledge the clarity of the paper. Given this disagreement, we believe that it's a bit unfair to recommend a rejection of the paper solely based on this criterion. We’ve provided detailed responses to each reviewer and made substantial revisions (added one page) to the manuscript for clarification, including expanding the related works section, adding an extra appendix section of option visualizations, cleaning up notations, and further emphasizing our contributions . By doing so, we hope that the reviewers’ concerns will be largely addressed.

~~(Our detailed responses below are based on the updated version, dated Nov 12, 2020.)~~

UPDATE (Nov 19, 2020): We have now included an extra experiment comparing task performance across all four environments with different option lengths and a visualization of the trajectory distribution induced by options of different lengths in two of our environments. These figures correspond to Figures 7 and 8 in the appendix, and new analysis of these results has been added in Appendix Section C. This further addresses the concerns raised by AnonReviewers 1 and 3 regarding experiments for varied option lengths. We have updated the responses to those reviews accordingly.

---

### Decision · Program_Chairs · 2021-01-07
**Final Decision**

**Decision:**

Accept (Poster)

**Comment:**

This paper presents an approach to hierarchical RL which automatically learns intrinsic task-agnostic options. The approach involves a two-level hierarchy, with policies learned by lower-layer Workers and selected by a higher-layer Scheduler. The approach is evaluated on four complex tasks and is shown to outperform existing methods.
There were initial concerns with this paper around clarity of a number of points. These included the contributions of this work and questions around the experimental results, such as  discussing the learned options themselves. The authors provided extensive responses to these concerns, and updated the paper accordingly, including addition results and analysis. I believe the paper is now much clearer with interesting contributions.